# Identifying Infant Body Position from Inertial Sensors with Machine Learning: Which Parameters Matter?

**DOI:** 10.3390/s24237809

**Published:** 2024-12-06

**Authors:** Joanna Duda-Goławska, Aleksander Rogowski, Zuzanna Laudańska, Jarosław Żygierewicz, Przemysław Tomalski

**Affiliations:** 1Neurocognitive Development Lab, Institute of Psychology, Polish Academy of Sciences, ul. Jaracza 1, 00-378 Warsaw, Poland; zlaudanska@psych.pan.pl; 2Faculty of Physics, University of Warsaw, ul. Pasteura 5, 02-093 Warsaw, Poland; j.zygierewicz@uw.edu.pl

**Keywords:** inertial motion sensors, human activity recognition, explainable machine learning

## Abstract

The efficient classification of body position is crucial for monitoring infants’ motor development. It may fast-track the early detection of developmental issues related not only to the acquisition of motor milestones but also to postural stability and movement patterns. In turn, this may facilitate and enhance opportunities for early intervention that are crucial for promoting healthy growth and development. The manual classification of human body position based on video recordings is labour-intensive, leading to the adoption of Inertial Motion Unit (IMU) sensors. IMUs measure acceleration, angular velocity, and magnetic field intensity, enabling the automated classification of body position. Many research teams are currently employing supervised machine learning classifiers that utilise hand-crafted features for data segment classification. In this study, we used a longitudinal dataset of IMU recordings made in the lab in three different play activities of infants aged 4–12 months. The classification was conducted based on manually annotated video recordings. We found superior performance of the CatBoost Classifier over the Random Forest Classifier in the task of classifying five positions based on IMU sensor data from infants, yielding excellent classification accuracy of the Supine (97.7%), Sitting (93.5%), and Prone (89.9%) positions. Moreover, using data ablation experiments and analysing the SHAP (SHapley Additive exPlanations) values, the study assessed the importance of various groups of features from both the time and frequency domains. The results highlight that both accelerometer and magnetometer data, especially their statistical characteristics, are critical contributors to improving the accuracy of body position classification.

## 1. Introduction

The development of human motor behaviour involves learning through practice as infants improve their skills over time. Changes in motor behaviours take place across long periods of time (days, weeks, or months), and being able to capture such changes in detail across longer timeframes in multiple situations may help us discover developmental trajectories of emerging motor skills. Pioneering work by Bouten et al. [1] laid the groundwork by developing a device that combined a triaxial accelerometer with a data processing unit to measure body accelerations. This early work demonstrated the potential of accelerometers for capturing a broad range of physical activities. Najafi et al. [2] advanced the field by incorporating gyroscopes to monitor postural transitions, which facilitated the use of Inertial Motion Units (IMUs) in various applications, including fall detection [3,4], stair activity monitoring [5,6], and non-invasive physical movement monitoring using magnetic and inertial measurement units [7]. Additionally, IMUs have been employed for detailed gait analysis with integrated sensors [8]. Accelerometers are effective in capturing a range of movements, from walking and running to more subtle activities, while gyroscopes measure rotational movements essential for understanding body dynamics. They offer a comprehensive view of body orientation and movement dynamics when used in conjunction with magnetometers. Recent advancements in sensor technology and classification algorithms have further improved the possibilities for this research.

In recent years, thanks to the miniaturisation of IMUs, wearable technologies have proven valuable for capturing real-time data on early development, providing new insights into the neurodevelopmental profiles of typically and atypically developing infants. Given the portability, mobility, low hardware cost, small size, and low weight of new wearable technology, it has tremendous potential in infant studies in the laboratory, clinical settings and natural environments—including the acquisition of infants’ data without the presence of experimenters or clinicians. Wearables provide valuable insights into physical development and motor skills during infancy [9] and their potential effects on cognitive development [10]. The integration of accelerometers, gyroscopes, and magnetometers has enabled a detailed examination of physical activities and motor skills, providing a better understanding of developmental changes in movement patterns and body positions. For example, studies [11,12] have successfully classified different types of movement using accelerometer and gyroscope data.

A particularly important application of wearables in paediatric research is tracking the changes in body positions in infancy. Infant body positions influence how caregivers interact with their infants and the cognitive opportunities they provide [13]. Previous studies [14,15,16,17] have demonstrated that it is viable to identify and categorise multiple body positions in infants based on sensor data. Greenspan et al. [18] also showed that wearables could be feasible for measuring parent–infant positioning practices. However, it has yet to be determined how the number of sensors, the type of machine learning algorithms, and the choice of features affect classification outcomes in wearable devices used for infant monitoring.

In recent years, machine learning techniques have been increasingly applied to classify movement and body positions. Algorithms such as Support Vector Machines (SVMs) [19], Random Forests [14,20], and Convolutional Neural Networks (CNNs) [15,16] have shown promise in enhancing classification accuracy. For example, infant studies have used SVMs to achieve classification accuracies above 90% for activities like walking, running, and sitting [15,17]. However, these techniques, especially CNNs, often require large datasets and extensive training periods, which can limit their applicability in real-time or unsupervised environments.

Signal segmentation is a crucial stage in the process of classifying body position. Current research limitations include reliance on extended activity windows for classification, which may not accurately reflect real-time changes in body position. The length of activity windows used in position classification varies significantly, ranging from short intervals of approximately 0.1 s [21,22] to longer windows exceeding 8 s [23,24], depending on the number and placement of accelerometers and the specific activity being monitored. To capture movement dynamics, data can be partitioned into segments using activity-defined windows [25], event-defined windows [26], or sliding windows, which is the most commonly applied technique. While overlapping adjacent windows are sometimes acceptable for specific applications, they are less commonly employed [14]. Additionally, variations in sensor placement and calibration can introduce noise and affect data consistency.

To address these limitations, our study aimed to develop an automatic system for classifying infants’ body positions using data collected from accelerometers, magnetometers, and gyroscopes during three different brief activity windows. Each infant took part in play activities with the caregiver typical for their age range (book-sharing, playing with toys, and rattle-shaking). These activities allowed us to sample different kinds of movements and spontaneous changes in body position under semi-naturalistic conditions. This approach was designed to provide more granular and accurate results than previous studies. Using data from an existing longitudinal study of infants aged 4–12 months [27], we aimed to offer a more precise and practical method for monitoring and analysing infants’ motor development.

In addition to classifying body position, it is essential to explore the importance of different features within the IMU data and how each contributes to the accuracy of the classification. By utilising the established characteristics of the signal, we can designate feature groups through methods from the time-frequency domain [28,29], statistical analyses [14], or heuristic strategies [30]. By identifying which feature groups—such as signal variability, movement frequency, or the dependency between sensors—are most influential in distinguishing various positions, we can gain deeper insight into motor development. For instance, variations in signal frequency may be critical for differentiating crawling from walking, as each activity exhibits distinct movement patterns and characteristics. Understanding these differences can help us better interpret the mechanics of motor development in infants.

This analysis of feature importance not only enhances the interpretability of our model but also aids in selecting a tailored subset of features specifically designed for the activities we are focusing on. This optimises both the number and placement of sensors used, thereby improving classification algorithms. Furthermore, by identifying key features, we can extract them from different datasets to better understand how these datasets vary—a point we will explore further in the discussion.

In a recent review [31], various machine learning methods were evaluated for their accuracy in position classification. For example, Supine shows accuracy rates between 87 and 97%, while Prone ranges from 67 to 98%, and activities such as pivoting and crawling commando show lower accuracies of 62–66% and 60%, respectively. Upright positions, including standing, walking, and running, exhibit a wide accuracy range from 9 to 100%.

However, these comparisons are problematic due to several factors. Differences in datasets, including variations in sample size, sensor types, and environmental conditions, significantly affect results. Additionally, the models used vary, skewing direct performance comparisons. Inconsistent sets of positions and participant age ranges across studies further complicate the reliability of these comparisons.

To address these challenges, we will explore the implications of dataset variability and model differences in our analysis.

## 2. Materials and Methods

### 2.1. Experimental Design

Data were collected from a longitudinal study of infant-caregiver dyads at four time points corresponding to the infants’ chronological ages of 4, 6, 9, and 12 months.

Families were briefed and provided informed consent upon arrival at the laboratory, where interactions were recorded in a carpeted play area. Infants and caregivers were fitted with motion sensors and head cameras, though their data are not reported here.

Each visit involved 6–7 activities, conducted in randomised order, with toys differing between younger (4–6 months) and older infants (9–12 months). This analysis focuses on three play activities: book-sharing, rattles, and an object play task (“manipulative task”), each lasting about 5 min (see Table 1).

The toy sets varied between the older and younger age groups to ensure a similar level of interest and alignment with their cognitive and motor skills. The dyads had full autonomy in their use of the toys, moved freely around the room, and did not receive any specific instructions with respect to how to structure the activity or which body position to take.

Interactions were captured using three remote-controlled HD CCTV [32] cameras and a high-grade cardioid microphone (Sennheiser e914 [33]) for synchronised audio–video recording.

#### 2.1.1. Participants

A total of 104 infant-parent dyads participated in the study [27]. Of these, 48 provided data at all four time points, while 83 missed one visit, primarily due to COVID-19 restrictions (see Table 2). We included in the analysis only 301 visits with manually annotated body positions. The mean ages of each time point were 4 months (ranges from 3.9 to 5.2 months), 6 months (ranges from 6 to 7.8 months), 9 months (ranges from 8.2 to 10.2 months), and 12 months (ranges from 11 to 14.5 months). For additional details, refer to Table 2.

#### 2.1.2. Equipment

Infants’ and caregivers’ body movements were recorded at 60 Hz using wearable motion trackers [34] (MTw Awinda [35]), which were wirelessly connected to an MTw Awinda receiver station [35] and synchronised in real time with MT Manager software version 2020.02 [35]. These are advanced devices that measure acceleration, angular velocity, and magnetic field strength. A total of twelve sensors were used, with two sensors placed on each of the infants’ arms, legs, head, and torso, as well as two sensors on the caregivers’ arms, head, and torso. However, this paper focuses solely on data from three pairs of sensors placed on the infants’ legs, torso, and arms (see Figure 1). The sensor placed on the head of the infant was removed often, resulting in its exclusion from the analysis.

A comprehensive description of the sensors’ operation can be found in [36].

#### 2.1.3. Coding of Infant Position

The infants’ body positions were manually annotated in ELAN 6.3 based on position protocol. Each video was viewed at 1× speed to identify episodes of postural changes, with the precise onset and offset times annotated frame-by-frame. Positions were classified based on the period between the first and last frames in which the infant was observed in that particular position according to the definitions in Table 3, adapted from Thurman and Corbetta [37]. Transition periods between positions were not assigned to any specific positions.

In the set of 14 annotated body positions, positions were divided into two categories: dynamic positions (such as walking or crawling) and static positions (such as lying down or standing). The dynamic positions were then mapped to their corresponding static positions, resulting in 5 distinct classes. These five classes were used for training the classifiers. The names of these classes and their corresponding positions are listed in Table 3. Figure 2 depicts the prevalence of different positions across time points.

### 2.2. Data Pre-Processing

The IMU sensor data, collected from an infant’s arms, torso, and ankles, were processed using custom MATLAB 2019b [38] scripts. Figure 3 illustrates sensor readings and manually assigned postural positions.

The analysis focused on the accelerometer, gyroscope, and magnetometer signals. The IMU tracking system, which measures user orientation wirelessly via WiFi, occasionally faced connectivity issues that resulted in data gaps. These gaps were primarily due to the IMU sensors’ internal characteristics and automatic changes in the sampling rate from 60 Hz to 40 Hz. To ensure the time series data remained consistent, missing values were interpolated using the ‘spline’ method with MATLAB’s interp1 function [38]. Additionally, when a lower sampling rate of 40 Hz was identified, the signal was resampled to 60 Hz using MATLAB’s [38] resample function. To mitigate artefacts stemming from instances where the infant displaced the sensor and was stationary on the ground, the acceleration magnitude variance was computed in 0.1-s windows. Data signals with variance below 1 × 10^−6^ m/s^2^ were identified as artefacts and subsequently excluded from the analysis. The dataset consisted of sensor readings from accelerometers (Acc), gyroscopes (Gyro), and magnetometers (Mag). From accelerometers, the data encompassed primary accelerations for 6 locations and three orthogonal axes. Subsequently, accelerometer data were High- and Low-Pass-filtered (HP and LP). The resulting AC and DC acceleration data groups were integrated with the original sensor data. Additionally, the magnitudes (Euclidean norms, Norm) of both the original (X, Y, Z) and newly derived acceleration sensor data were included in the analysis.

Gyroscopes provided primary angular velocity values, and magnetometers contributed primary magnetic field strength values to the dataset. Initially, signals were selected for further analysis, considering using various toys to engage infants. Despite being in the same body position, movements during play with different toys, both within and across tasks, varied significantly. The hands emerged as a major source of signal variability across tasks, leading to the exclusion of hand sensor data in subsequent analyses.

For details on all signals, see Appendix A, Section 2.

#### Synchronisation of Movement and Audio-Video Data

At the beginning of each task, the caregiver was instructed to clap five times to provide an audible synchronisation cue to align annotations with the IMU sensors. Subsequently, all claps were manually annotated using both audio and video data, which had been pre-synchronised. Next, the signal representing the Euclidean norm of the three-dimensional acceleration vector of caregiver’s arm IMUs was calculated using Equation A1. Then, the delay between the signal and audio annotations was calculated based on the caregiver ‘claps’ detected in averaged Acc wearable signals from both hands and applied to synchronise the signals from IMU sensors to the audio–video stream.

### 2.3. Class Selection and Parameter Extraction

#### 2.3.1. Identifying Relevant Classes

In the dataset comprising 14 classes, as determined by annotators coding video recordings from the study, body positions were categorised into dynamic (e.g., walking, crawling) and static (e.g., lying, standing) subsets. A subsequent mapping process combined dynamic positions with their corresponding static counterparts, resulting in a refined set of 5 classes for classifier training.

#### 2.3.2. Extracting Features and Composition of Feature Groups

To extract parameters, IMU signals were analysed using 2-s sliding windows with a 1-s overlap between consecutive windows. Each window was examined only if at least 75% of its samples were consistently assigned the same position label. Figure 4 illustrates the process of analysing these overlapping windows using a manually annotated data segment.

The features extracted from IMU sensors are categorised into five main groups (Figure 5), each offering unique insights into the data. Frequency-domain features encompass the properties of energy, entropy, centroid, bandwidth, and maximal frequency to capture the fundamental signal properties. The remaining features are calculated in the time domain. Statistical features describe the data distribution, including the metrics of mean, standard deviation, median, skewness, kurtosis, and key quantiles, along with minimum and maximum values. Summary features provide aggregated information by summing data across axes and locations. Difference features highlight variations within the data by calculating mean differences between axes and locations. Finally, correlations examine the relationships between different features to reveal their interactions and dependencies.

To enhance the model’s ability to infer orientation, we combined Roll and Pitch features [39,40] with the low-pass filtered accelerometer signal. These features provide detailed information on the device’s tilt and angular position, significantly influencing position classification. Integrating these specific features allows the model to capture subtle variations in orientation that may be crucial for distinguishing between different positions.

These feature groups collectively offer insights into the relationships and significance of various IMU signal properties in position classification tasks using IMU sensor data. The complete list of features and their formulas can be found in Appendix A, Section 3.

### 2.4. Classifier Selection

In this study, we chose two different types of classifiers for body position classification from IMU sensor data. The first classifier is Random Forest (RF), commonly used in IMU-based position studies (e.g., [14,41,42,43]) as a standard model, and it thus served as a reference. The Random Forest algorithm is an ensemble method that constructs multiple decision trees, where the variability of individual tree predictions is reduced by introducing randomness during the classifier construction. This randomness is achieved through bootstrapping the training set and selecting a random subset of features at each split within the trees. The final prediction of the ensemble is obtained by averaging the predictions of the individual trees. In this work, we utilised the RandomForestClassifier implementation from the scikit-learn library [44] with the following parameters: [n_estimators = 1000, max_depth = 6, class_weight = ‘balanced’].

The second classifier we used is a gradient-boosted model, specifically the CatBoostClassifier from the CatBoost library [45]. This model constructs multiple shallow decision trees, iteratively fitting each tree to the residual errors of the preceding trees. To further reduce the risk of overfitting, the CatBoost algorithm randomly shuffles the features at each tree level. The tree predictions are then aggregated to enhance classification accuracy and improve the model’s generalisation capabilities. Gradient-boosted classifiers like CatBoost have shown superior performance in various datasets, especially in bioinformatics applications, as noted by Olson et al. [46].

### 2.5. Classifier Performance Evaluation

The models were trained in a 5-fold cross-validation procedure (see Figure A1) to estimate their average performance and standard error. Folds were constructed so that recordings from a given infant are only in one fold to prevent data leakage.

We evaluated our models utilising the F1 score as it is one of the most popular methods and considers both precision and recall as a harmonic mean. F1 score reaches its best value at 1 and worst score at 0. The relative contribution of precision and recall to the F1 score are equal. In the multiclass problem at hand, we evaluated the F1 score, treating our data as a collection of binary problems, one for each position vs. other positions.

The confusion matrix displays the counts of correct classification (on the diagonal) and mis-classifications (off-diagonal). Analysing these values allows for a detailed assessment of the model’s performance, revealing its strengths and weaknesses in classifying instances of various classes. This breakdown provides additional insight beyond the F1 score, offering a more transparent and more nuanced understanding of the model’s classification behaviour and biases.

### 2.6. Importance of Different Features Groups

#### 2.6.1. Ablation Experiments

The importance of feature groups was evaluated using ablation experiments and F1 score analysis. We performed two types of ablation experiments: in the first, individual feature groups were removed, and in the second, models were trained using only one specific feature group at a time. For both experiments, we assessed the impact on model performance by analysing the resulting changes in the F1 score values.

#### 2.6.2. SHAP Values

To assess feature group importance, we utilised SHAP (SHapley Additive exPlanations) values, which offer a game-theoretic approach to explain machine learning model outputs. SHAP values reveal how each feature influences predictions, their relative significance, and the interplay between features.

We used TreeExplainer from the SHAP library [47] and evaluated feature importance through the mean and sum of |SHAP| values (of absolute values), where the averaging or summation was performed within the feature groups of a given signal (Figure 1). Mean |SHAP| values show the average contribution of each feature across different folds, helping us understand their general impact and compare their relative importance. This method also highlights feature stability and provides insights into uncertainty through associated error measurements.

On the other hand, the sum of |SHAP| values captures the total contribution of each feature group across all folds, reflecting the aggregate impact of features on model predictions. This approach allows us to assess the cumulative influence of feature groups and compare their overall effects. While it does not directly reveal feature interactions, it highlights which feature groups have the most significant total impact.

### 2.7. Correlation Between Annotated and Predicted Time in Position

To assess how accurately the model’s predictions align with the true annotated data, we examined the correlation between annotated and predicted time for a position. For each position, we calculated the temporal sums of windows labelled and assigned by the classifier to it. Then, they were divided by the overall time of the respective study and multiplied by 100 to obtain the percentage share of a given position in the entire study. A high correlation within position indicates that the model effectively captures the time spent in different positions, which is crucial for tasks such as position monitoring and assessment.

## 3. Results

### 3.1. Classifier Comparison

#### Sensor Placement and Classifier Comparison

Figure 6 shows the results of the statistical comparison of three CatBoost and Random Forest classifiers, each trained and evaluated on various combinations of sensor data. CatBoost consistently demonstrated superior performance compared to Random Forest across all three models: Trunk; Trunk and Legs; and Trunk, Legs, and Arms. This consistent advantage suggests that CatBoost’s ability to handle categorical features and its robust gradient-boosting framework contribute significantly to its effectiveness in this context.

Sensor placement is pivotal for effective position classification. Our findings indicate that the optimal configuration involves using sensors on the trunk and legs, which yields the highest F1 scores. This result may be largely attributed to the engagement of the hands in object-related actions. As expected, relying solely on trunk sensors results in reduced F1 scores, especially for Upright. Notably, incorporating sensors on the arms does not enhance accuracy when combined with trunk and leg sensors. As illustrated in Figure 6, the averaged F1 scores across five folds demonstrate that the Trunk and Legs configuration outperforms the others, while the addition of arm sensors provides no discernible advantage. For more details, see Appendix A
Figure A2.

Taking the above results into account further, we will focus on analysing the CatBoost models with Trunk and Legs sensors as input analysis.

### 3.2. CatBoost Performance Evaluation: Trunk and Legs

#### 3.2.1. Confusion Matrices

The confusion matrix (Figure 7) reveals that the model excels at predicting Supine and Sitting positions, with both categories achieving over 90% accuracy—a performance that highlights the model’s reliability. Notably, Sitting is identified accurately most frequently, although it occasionally gets misclassified as Hands and Knees and Prone.

Predictions for Upright are also highly accurate, with minimal misclassification into other categories. However, the model demonstrates less accuracy for the Prone position, which is often misclassified as Sitting and Supine.

In contrast, the Hands and Knees position shows moderate accuracy, but it is frequently confused with Sitting and Prone. This trend suggests that the least stable position generates the most errors, which aligns with expectations, as it tends to transition into movement or shift to a more stable position.

#### 3.2.2. Ablation Experiments

The results presented in this section reflect the percentage change in model performance during two types of ablation experiments using CatBoost to observe the effect on classification accuracy for various infant positions: Sitting, Upright, Supine, Prone, and Hands and Knees and with different feature groups, including Statistical, Frequency, Summary, Difference, and Correlation.

##### One Feature Group at a Time vs. All Feature Groups

The results depicted in Figure 8 reveal how the performance of a model, measured through F1 scores, changes when using only one feature group at a time compared to a model using all available features.

For Statistical features, the differences are relatively small, with the model showing slight improvements or declines depending on the position. The upright position, in particular, shows a modest increase, while other positions experience minor decreases or nearly no change.

In contrast, using only Frequency features consistently leads to a drop in F1 scores across all positions. The model performs significantly worse when only frequency-related features are used, particularly for more complex positions like Hands and Knees or Supine. This suggests that frequency information alone cannot capture the full complexity of these positions.

The Summary features follow a similar trend as Frequency features, with a negative impact on performance, though the changes are not as drastic as those seen for Frequency features. Positions like Supine and Prone are particularly affected, where the model struggles to maintain accuracy when relying solely on summary data.

Restricting the input to Difference features produces some of the most significant declines in performance, especially for challenging positions like Hands and Knees. This indicates that the Difference-based features, though valuable in some contexts, may not capture the full scope of the data needed to classify positions accurately.

Finally, limiting the input solely to the Correlation features shows a consistent negative impact across all positions, with Upright and Hands and Knees again standing out as the most affected. The reliance on correlation alone seems insufficient to maintain high performance, suggesting that it plays a supplementary rather than a primary role in body position classification.

##### Excluding a Single Feature Group

For the Sitting position, omitting Statistical features leads to a slight reduction in the F1 score, while excluding Frequency has minimal impact. Notably, removing the Summary results in a noticeable decline, whereas omitting the Difference yields a marginal improvement.

In the Upright position, excluding Statistical and Frequency features significantly lowers the F1 score, with Summary causing a considerable drop. Conversely, omitting Difference enhances the score.

For the Supine position, excluding Statistical features results in a marked reduction, while removing the Summary leads to an improvement. The impact of omitting other features is minimal.

In the Prone position, excluding Statistical features causes a moderate drop, while the absence of Frequency provides a slight improvement. The Summary removal results in a small gain.

Finally, for the Hands and Knees position, excluding Statistical features leads to a significant drop, with Frequency also causing a considerable decrease. However, omitting Difference slightly enhances the score.

Overall, the most notable impacts arise from the exclusion of Statistical and Summary features across various positions. The results are shown in Figure 9.

#### 3.2.3. SHAP Values

The analysis of SHAP values reveals a compelling hierarchy in the influence of various feature groups on the model’s predictions. Statistical features emerge as the most important, commanding the highest total |SHAP| values. This finding underscores their dominant role in shaping model outcomes. Difference features follow closely, contributing significantly, though to a somewhat lesser extent.

Alternatively, the Correlation and Summary features exhibit a more subdued influence, while the Frequency group ranks lowest in impact. When considering sensor contributions, accelerometer signals are the most critical, playing a pivotal role in enhancing model performance, followed closely by the magnetometer signals. On the other hand, gyroscope signals contribute the least to the model’s predictive capacity.

The distribution pattern of the most significant sum of |SHAP| values remains consistent across all examined positions. The most pronounced influence is observed in the low-pass filtered data from the accelerometer’s Statistical features. For further details, refer to Figure 10.

Mean |SHAP| values (see Figure 11) serve as a valuable tool for comparing the typical effects of features, especially given the varying sizes of feature groups. The signals from Pitch and Roll emerge as significant, particularly when combined with Difference, Statistical, and Summary features.

Notably, similar to the pattern observed with summed |SHAP| values, the distribution of the most significant mean |SHAP| values remains consistent across all examined positions. The combination of Pitch and Summary features exhibits the strongest influence, underscoring their critical role in enhancing the model’s overall performance.

#### 3.2.4. Correlation Between Manually Annotated and Predicted Time in a Given Body Position

In Figure 12, we illustrate the correlation between the annotated, actual, and predicted time spent in five distinct positions across study sessions for five positions. The strongest correlations occur for Supine (r = 0.99) and Sitting (r = 0.93). Prone and Upright also exhibit very high correlations (both r = 0.92). The lowest predicted time spent in a single position is for Hands and Knees, with a correlation of r = 0.76.

Among the three activities (tasks), the weakest correlation is for the Manipulative task, followed by the Book-sharing task. In contrast, the Rattles task (where r ranges from 0.89 to 0.99) demonstrates the most accurate predictions for time spent in each position using CatBoost. Notably, Rattles is the only task that contributed data from all four time points, providing the most comprehensive dataset to fit correlations between annotated data and machine learning predictions. However, it is essential to note that not all tasks and time points are accompanied by corresponding annotation data. Figure A3 shows the correlation between annotated and predicted time for data merged from all three tasks.

## 4. Discussion

In this study, we developed an automatic system for classifying infants’ body positions during naturalistic interactions with caregivers across the first year of life using data collected with Inertial Motion Units consisting of accelerometers, magnetometers, and gyroscopes. The classification was conducted on the basis of manually annotated video recordings. The body position classification from the IMU sensor data was conducted using two different types of classifiers: Random Forest Classifier and CatBoost Classifier.

We adopted an approach that employs 2-s activity windows with a 1-s overlap, as recommended by Banos et al. [48], which is relatively short compared to other studies [31]. This methodology enhances the accuracy of our results. We evaluated our classifier using three pairs of sensors placed on different body parts: Trunk, Legs, and Arms. Initially, we tested these sensor pairs, and subsequently, we analysed how different configurations of sensor pairs influenced classification accuracy using the following subsets: Trunk; Trunk and Legs; and Trunk, Legs, and Arms. The most promising results came from the Trunk and Legs configuration, likely due to the involvement of arms in task-related activities and object manipulation, regardless of the body position.

Although Random Forest is a well-established classification method, the superior performance of CatBoost in this study highlights the advantages of leveraging advanced algorithms that can better capture complex relationships in the data. CatBoost’s ability to handle nonlinear interactions between features made it better suited for this task, especially in capturing subtle distinctions across positions. This comparison highlights the potential of more sophisticated models to improve classification accuracy.

One of our key goals was to understand which parameters or feature groups contribute most to accurate position classification. Our findings indicate that certain feature groups are more effective at capturing specific positions. However, the model performs best when all feature groups are used together, highlighting that combining different types of hand-crafted features is essential for achieving optimal accuracy across various positions. Excluding features such as Difference, Frequency, and Correlation significantly impacts the classification of Prone and Supine. Interestingly, although Hands and Knees is a more dynamic position than Prone and Supine, the removal of these features still negatively affects its classification. Statistical and Summary features have a smaller impact, primarily influencing stable positions like Sitting and Upright. Exclusions of feature groups affect F1 scores differently across positions. The Upright position, which requires the most features, is particularly susceptible to exclusions, showing notable performance drops when omitting Statistical and Frequency features. Removing the Summary feature generally leads to a significant decrease in performance, while excluding Difference features often results in modest improvements. The absence of Correlation features typically causes a slight decline.

The analysis of |SHAP| values reveals a clear hierarchy among feature groups, with statistical features emerging as the most influential in shaping model predictions. Their high total sum |SHAP| values indicate a dominant role in model performance, closely followed by Difference features. In contrast, Correlation and Summary features contribute less, while the Frequency group has the least impact.

Among sensor contributions, accelerometer signals prove to be the most critical for enhancing model accuracy, with magnetometer signals also playing a role. Gyroscope signals, however, contribute minimally to predictive capacity. The distribution pattern of the most significant summed |SHAP| values remains consistent across all examined positions, particularly highlighting the strong influence of low-pass filtered data from the accelerometer’s statistical features. Focusing on static position is particularly important in contexts where infants maintain specific positions to manipulate objects effectively in task-oriented experiments. However, when distinguishing between dynamic positions, the gyroscope’s role becomes essential, as it provides real-time data on rotational movements and angular velocities, significantly enhancing classification accuracy in various dynamic scenarios.

Moreover, mean |SHAP| values provide valuable insights into the typical effects of features, especially given the uneven sizes of feature groups. In this sense, signals from Pitch and Roll are particularly significant, especially when combined with Difference, Statistical, and Summary features. The combination of Pitch and Summary features demonstrates the strongest influence, further emphasising their essential role in enhancing the model’s overall performance.

The Hands and Knees position, characterised by a limited number of instances, primarily functions as a transitional position within infants’ movement patterns, resulting in lower classification scores. However, the predicted time in this position still shows a good correlation with the actual time spent in this position. Feature selection plays a crucial role in this process. We incorporated Roll and Pitch angles from the accelerometer as intuitive parameters, significantly enhancing the accuracy of our automatic annotations.

Manual annotation offers frame-by-frame precision, achieving an accuracy of 0.04 s per frame at 25 frames per second. In our case, automatic annotation involves analysing signals in windows of up to 2 s with a 1-s shift, which can affect accuracy. The length of the time window for analysis directly impacts classification quality. Shorter windows may be more sensitive to positional changes but could result in a loss of contextual information. Conversely, longer windows might capture context better but are likely less precise regarding dynamic changes.

Determining whether to train the classifier using the entire activity window or a portion, such as 75%, is essential for optimising performance. Different activities influence accelerometer signals, leading to varied patterns of acceleration and variability. By adjusting the window length and shift interval, we can enhance the accuracy of automatic classification, facilitating more detailed analysis and statistical evaluation.

For static positions, using the longest window length proved most effective, allowing us to identify the Sitting, Prone, and Upright positions with over 90% accuracy.

Our results significantly surpass those reported in a recent review, which indicated wide variability in accuracy across different positions. This underscores the effectiveness of our methodology in achieving reliable position classification in a challenging environment.

It is also worth noting that dynamic positions, such as walking or running, tend to exhibit lower accuracy rates compared to static positions. This variability is expected, given the complexity and range of movements involved in dynamic actions, making consistent and accurate classification more challenging. To further refine our system, developing a classifier that distinguishes between static and dynamic positions would be beneficial. This could improve precision, as different classification strategies might be required for various positions.

Sensor placement also significantly impacts classifier accuracy. For instance, placing sensors on the wrists—typically involved in fine motor activities—can provide data specific to small-scale movements. However, if the classifier does not account for these movements, it may misinterpret the data, affecting overall position classification.

Furthermore, the fact that the classification was conducted jointly on data collected across a wide range of ages (4, 6, 9, and 12 months) and types of play facilitates the application of this methodology to infants with developmental difficulties. By concentrating on more universal patterns of behaviour regardless of age, researchers can more effectively capture diverse postural behaviours and movements of infants, who often do not adhere to typical developmental timelines. This flexibility enhances the relevance of the findings and allows for more tailored interventions that meet individual needs. Infants with motor delay display less mature positions than typically developing infants, which may result in fewer learning opportunities [13]. However, whether such differences could also be observed on longer time scales and during more naturalistic settings is unknown. Previous research highlighted the potential use of wearable motion tracking in monitoring infants’ physical activity levels [49], very early assessment of infant neurodevelopmental status in low-resource settings [50], and developing rehabilitation for infants with cerebral palsy [51]. Automatic classification of positions based on wearable motion trackers—potentially combined with other types of sensors, as demonstrated for general movement assessment by Kulvicius et al. [52]—can result in more personalised approaches to early diagnosis and intervention.

Although this study focused on classifying infant body positions using wearable sensor data, it is worth noting that video-based approaches can also be valuable tools in similar analyses. However, integrating video data presents certain challenges, such as the need for manual annotation and higher computational and time demands, especially when using algorithms like Tracking Learning Detection (TLD), introduced by Kalal et al. [53], which is designed for robust, long-term object tracking in video sequences [54]. In this study, we opted for an IMU-based approach because it offers high precision and enhances the ecological validity of research on natural interactions, where implementing video may be more challenging in a non-invasive manner. Nonetheless, video-based approaches can serve as a valuable complement, particularly when compared with sensor data. Through a reference validation method, this can enable the even more precise monitoring of infant body positions.

Our study focuses on classifying infants’ positions using a specific dataset and methodology, which may introduce certain biases, even under controlled conditions. While maintaining these conditions is crucial, having clean and well-prepared training datasets may be more advantageous.

Finally, the adequacy and reliability of manual annotation protocol is critical. Different research aims may require a more fine-grained categorisation of infant body position and locomotion.

## 5. Conclusions

Our study has significant scientific and practical implications for monitoring infants’ motor development. By employing shorter activity windows and incorporating frequency domain parameters, we have enhanced the accuracy of position classification. The superior performance of advanced algorithms, such as CatBoost, underscores the importance of sophisticated techniques in capturing complex relationships within the data.

Our findings indicate that statistical features, particularly those derived from accelerometer data, are crucial for differentiating body positions. Additionally, magnetometer data have proven to be useful in enhancing the accuracy of this differentiation. This capability facilitates the early detection of developmental delays, enabling timely interventions that can greatly benefit at-risk infants.

Furthermore, the integration of orientation-related data, including Roll and Pitch signals, contributes to improved classification accuracy. The insights garnered from this study support the development of wearable motion-tracking systems, which can enhance monitoring and early diagnosis in both community and clinical settings.

Future research should focus on validating these findings using different datasets collected under varied conditions and exploring additional sensor integrations to refine and expand the capabilities of position classification systems. By doing so, we can further advance our understanding of infant motor development and improve outcomes for those in need of intervention.

## Figures and Tables

**Figure 1 sensors-24-07809-f001:**
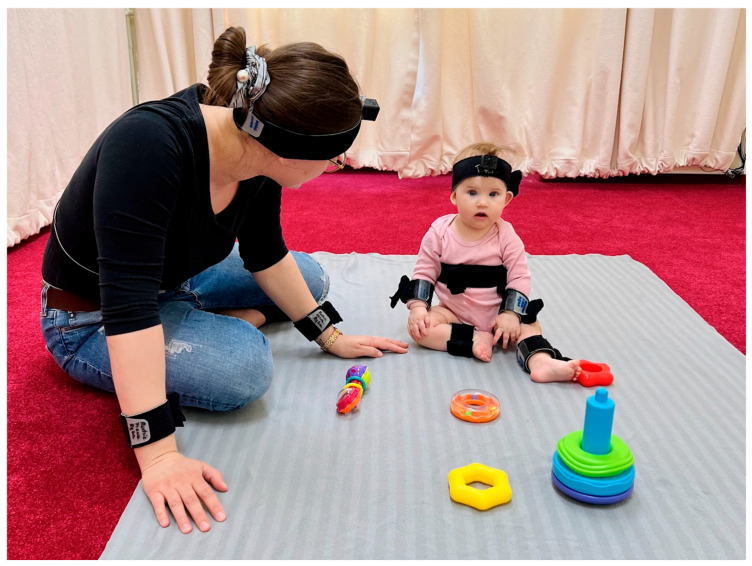
Placement of all sensors on the infant and caregiver. The Developmental Neurocognition Laboratory Babylab provided the photos at the Institute of Psychology, Polish Academy of Sciences, with written consent from the caregiver for publication.

**Figure 2 sensors-24-07809-f002:**
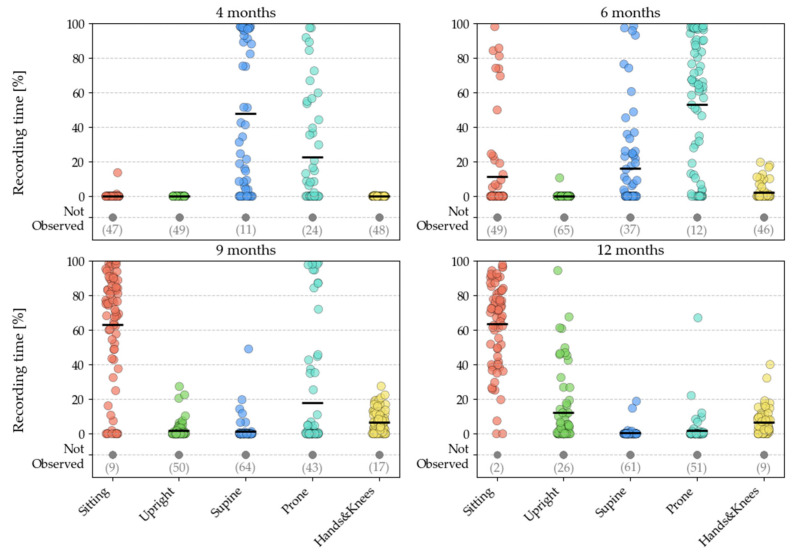
Percentage of total visit duration spent in various positions for each infant at the 4-, 6-, 9-, and 12-month time points. Each coloured dot represents the contribution of an individual infant. The numbers in brackets at the bottom of each plot indicate the number of infants that did not show a given position at a given time point. These plots illustrate changes in the distribution of postural behaviours as infants develop their gross motor skills over time.

**Figure 3 sensors-24-07809-f003:**
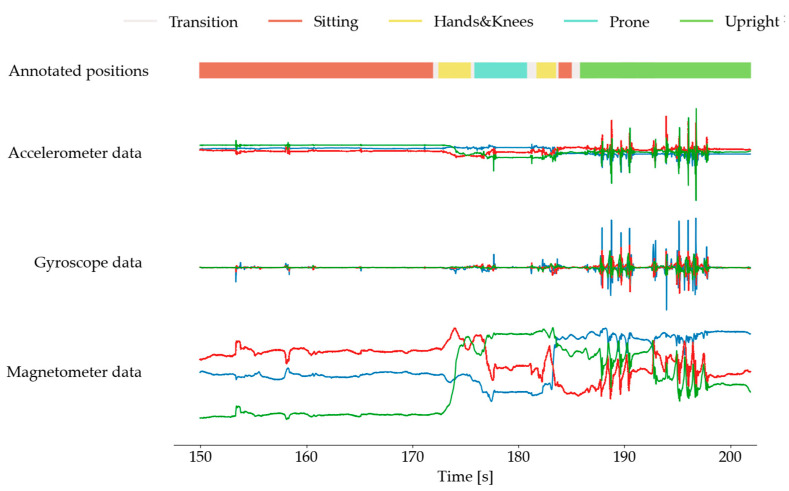
Illustration of sensor readings from tri-axial accelerometer, gyroscope, and magnetometer positioned on the left leg of a 12-month-old infant. The data are displayed across three axes (X, Y, Z) for each sensor type, showing variations in movement patterns associated with different body positions.

**Figure 4 sensors-24-07809-f004:**
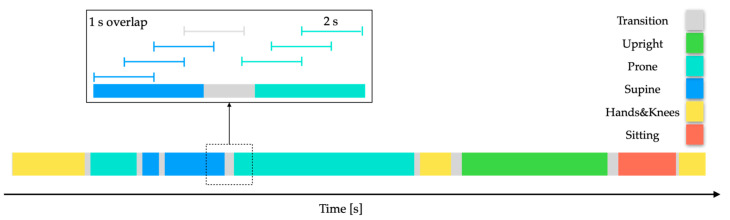
Sliding windows of 2 s with a 1-s overlap, showing assigned position fragments. Blue windows represent the Supine class, and teal windows indicate the Prone class. Windows where less than 75% of samples are consistently assigned the same position label were not assigned to any class.

**Figure 5 sensors-24-07809-f005:**
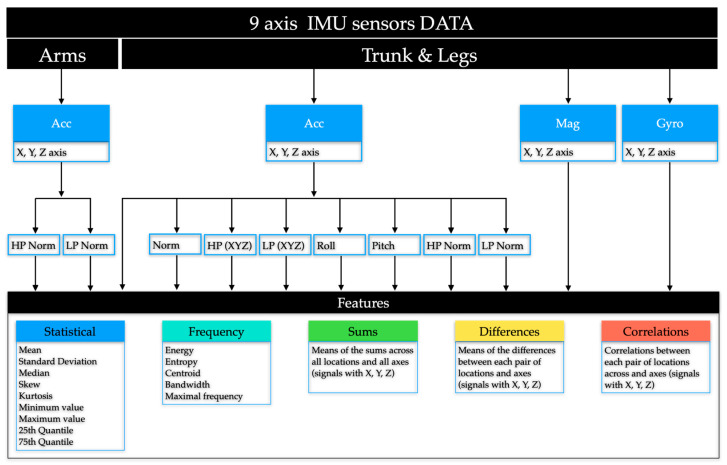
Composition of feature groups. Five distinct feature groups were extracted from each type of signal, including XYZ signals.

**Figure 6 sensors-24-07809-f006:**
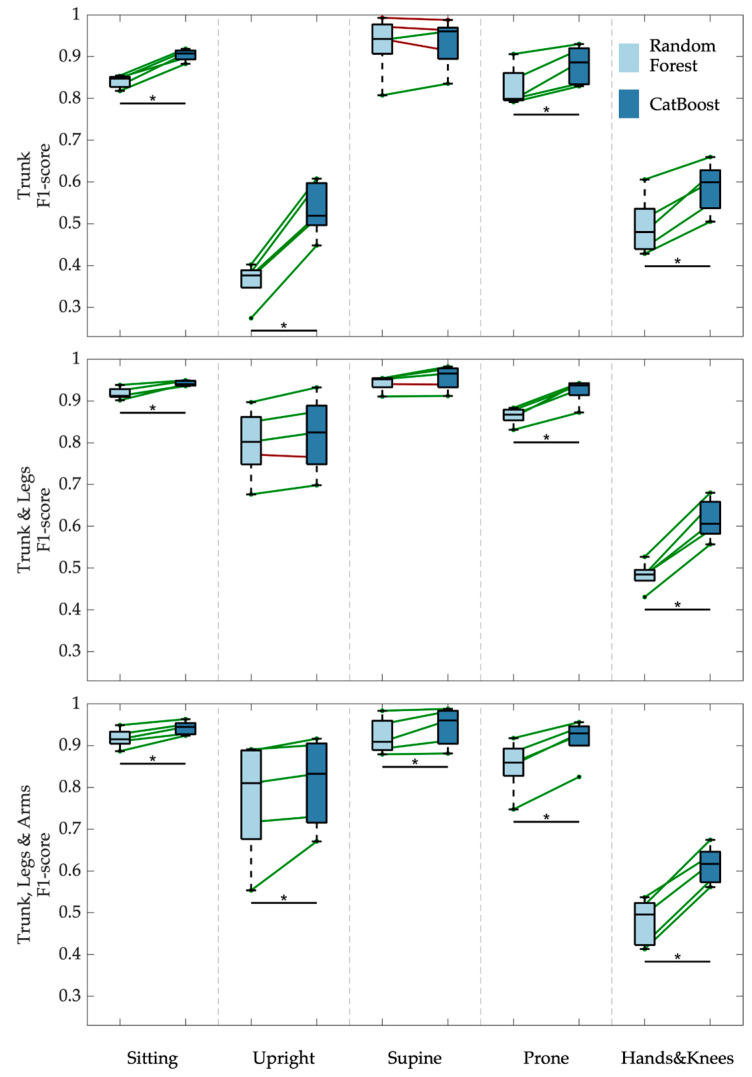
Comparison of F1 scores for Random Forest and CatBoost models across three sensors and five postures. Parallel coordinate plots highlight the performance differences, with green indicating superior performance by CatBoost and red indicating better performance by Random Forest. Each line corresponds to one fold in the cross-validation. Boxplots provide a visual summary of F1 score distributions for each model and posture. Statistical significance is indicated using FDR-adjusted *p*-values (the asterisks indicate the significance level * *p* < 0.05) derived from the Friedman test. The colour bar reflects the mean F1 scores for each posture, offering an overview of model performance across various conditions.

**Figure 7 sensors-24-07809-f007:**
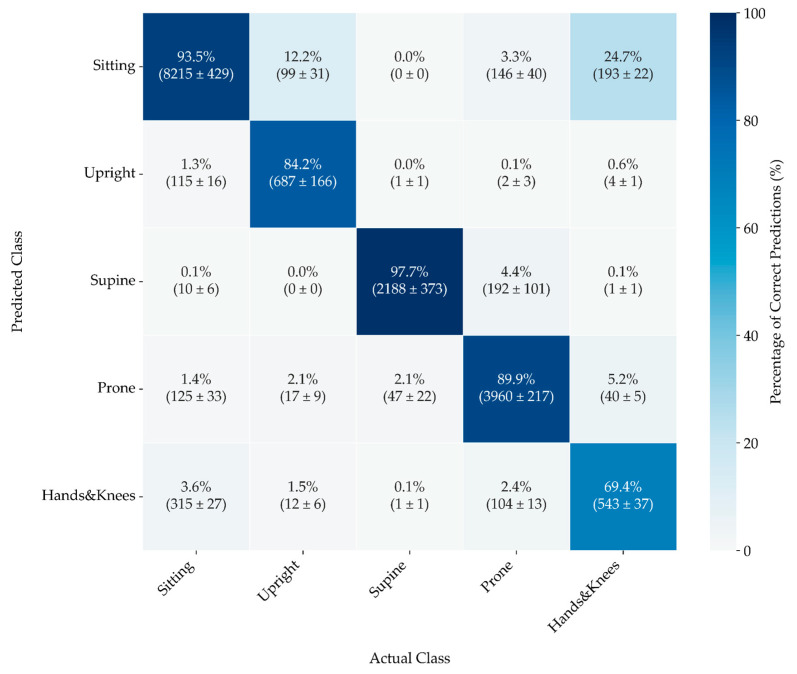
A confusion matrix was obtained based on the predictions of CatBoost classifiers trained on the combined set of parameters for two pairs of sensors: Trunk and Legs. The percentage of actual classifications is displayed at the top, with average counts across the five folds and their standard error included in parentheses. The colour scale corresponds to the percentage of the actual position, providing a visual representation of classification accuracy.

**Figure 8 sensors-24-07809-f008:**
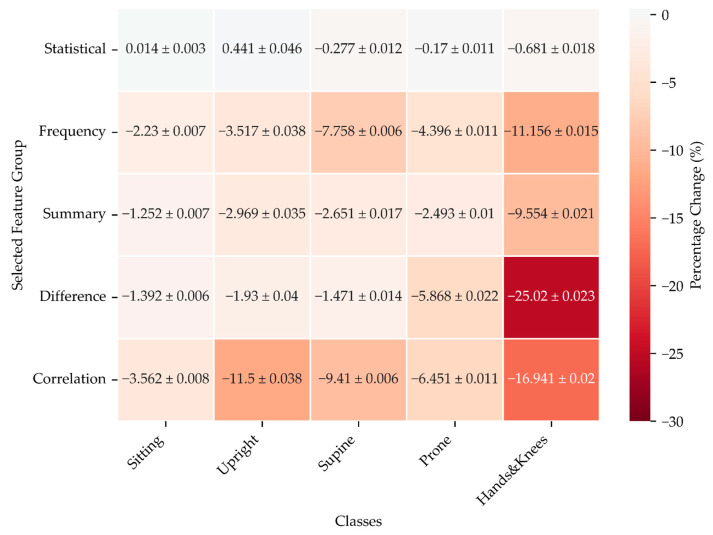
F1 score change for Catboost models using only one feature group at a time relative to models using all feature groups for two pairs of sensors: Trunk and Legs.

**Figure 9 sensors-24-07809-f009:**
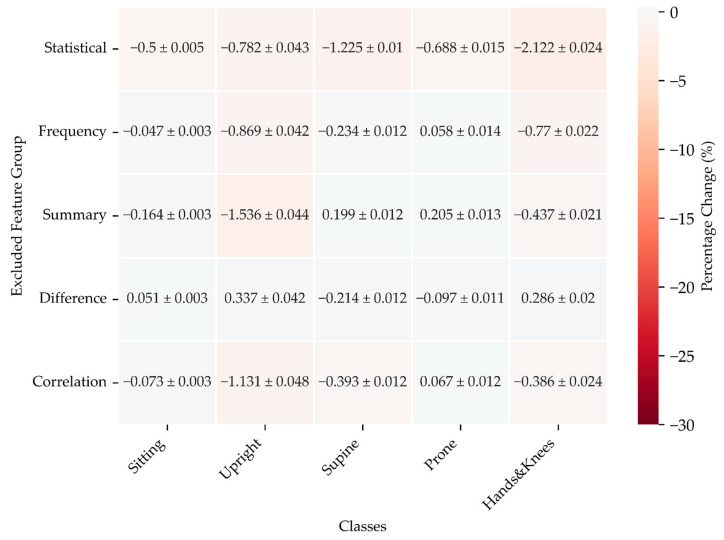
F1 score change for models excluding one feature group at a time relative to the CatBoost model using all feature groups for two pairs of sensors: Trunk and Legs.

**Figure 10 sensors-24-07809-f010:**
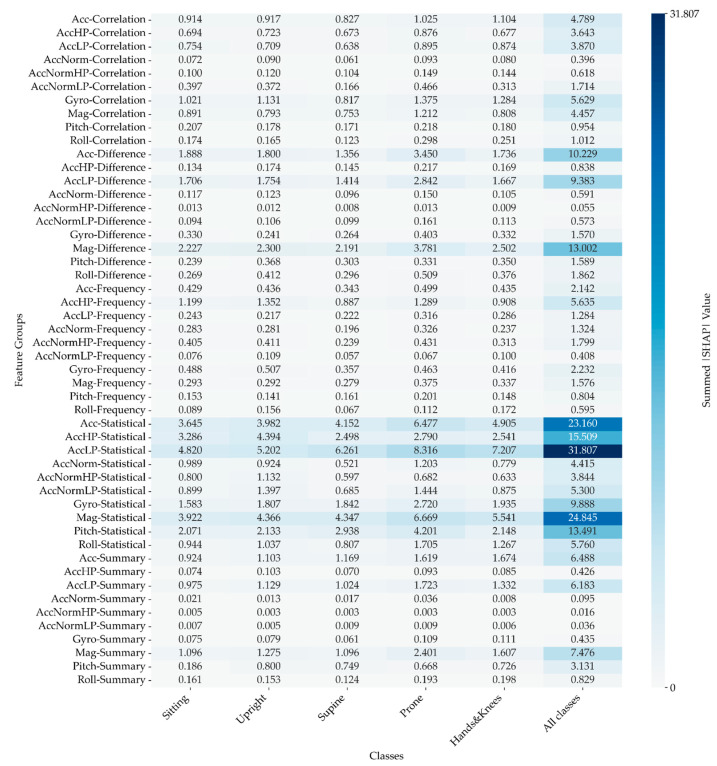
The sum of |SHAP| values across five folds illustrates the features with the highest total impact on the model, categorised by different signals and features. This figure highlights the most influential features in the model’s predictions, with larger-sum |SHAP| values indicating a more significant impact.

**Figure 11 sensors-24-07809-f011:**
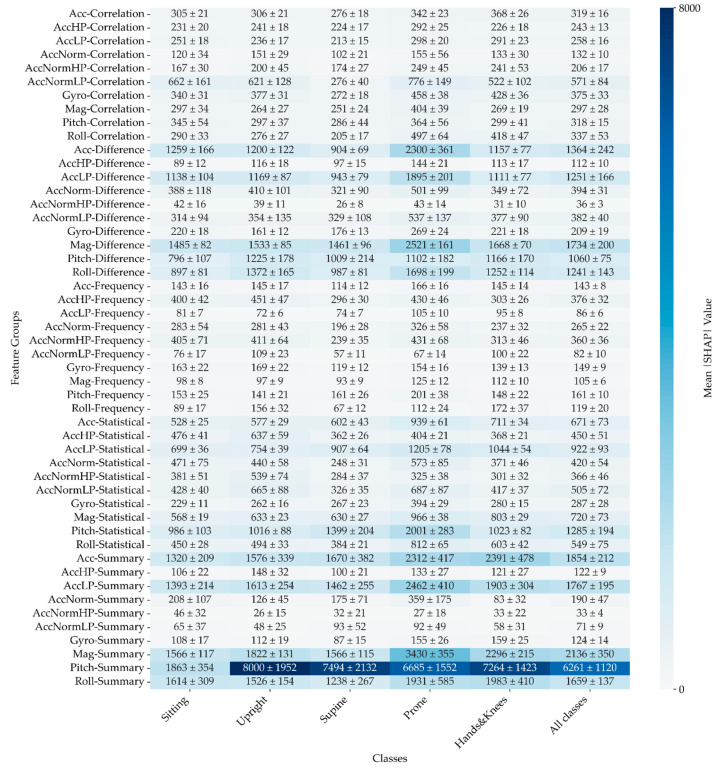
Mean |SHAP| values across five folds illustrate the features with the highest impact on the model, categorised by different signals and features. Presented values are multiplied by 1 × 10^5^ for clarity. This figure highlights the most influential features in the model’s predictions, with elevated SHAP values indicating a more significant impact.

**Figure 12 sensors-24-07809-f012:**
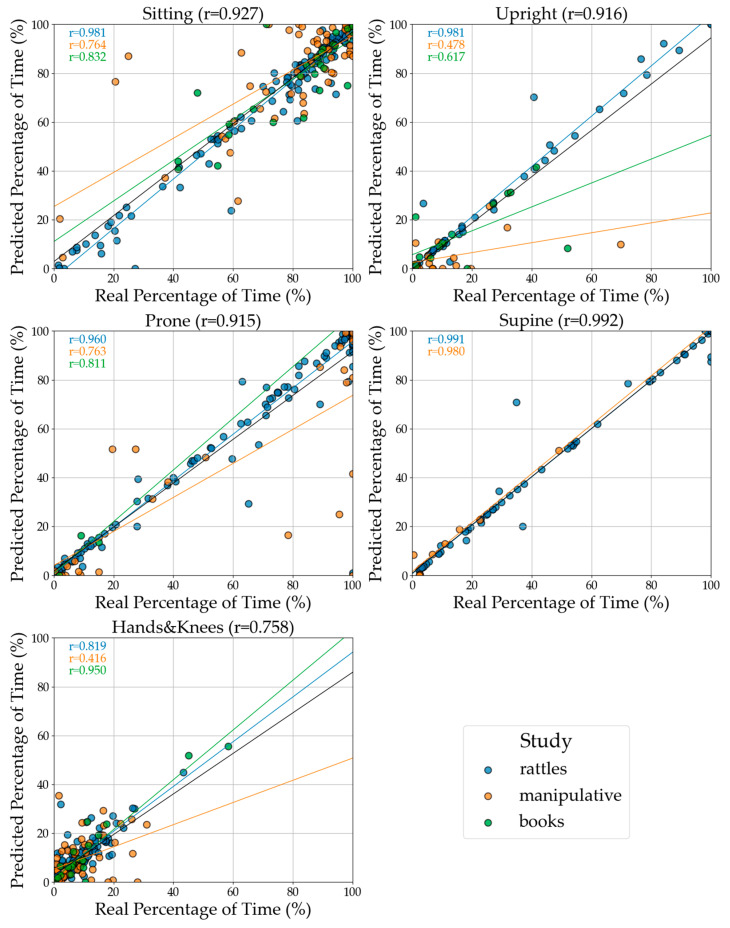
Illustration of the correlation between the annotated (actual) and predicted time spent in five distinct positions across study sessions. The x-axis represents the real percentage of time spent in each position, while the y-axis shows the corresponding predicted percentage. Each point in the plot corresponds to a specific infant in a position during one study session, with a line of best fit illustrating the correlation between the real and predicted values. The closer the points align with the line, the more accurate the predictions. Based on the CatBoost model for two pairs of sensors, Trunk and Legs.

**Table 1 sensors-24-07809-t001:** Toy sets in three activities (tasks) used in this study. The first row shows toys for infants aged 4 and 6 months, and the second row for infants aged 9 and 12 months. Photos are provided courtesy of the Neurocognitive Development Lab (Babylab) at the Institute of Psychology, Polish Academy of Sciences.

Time Point	Book-Sharing	Manipulative	Rattles
4–6 months	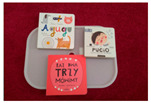	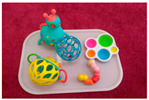	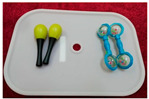
9–12 months	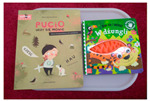	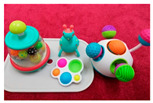	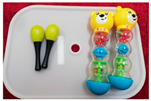

**Table 2 sensors-24-07809-t002:** The number of visits from infants participating at each age point during the selected task. Each infant could only contribute data once at each specific age and activity. The unique contribution of each infant at a given age prevents any overlap or duplication in the data.

Time Point	Rattles			Book-Sharing			Manipulative		
	Mean Age [Months]	SE [Months]	No. Visits	Mean Age [Months]	SE [Months]	No. Visits	Mean Age [Months]	SE [Months]	No. Visits
4 months	4.36	0.29	61	4.36	0.28	63	4.35	0.28	65
6 months	6.59	0.40	74	6.61	0.39	76	6.61	0.40	72
9 months	9.06	0.35	71	9.08	0.35	70	9.06	0.35	72
12 months	12.14	0.52	73	12.16	0.51	72	12.13	0.53	69

**Table 3 sensors-24-07809-t003:** Manually annotated static body positions, including their corresponding annotations and formal definitions.

Static Position	Annotated Position	Definition
Hands and Knees	hands and knees	The infant is on their hands and knees or feet with their stomach lifted off the ground and is not moving. Alternately, the infant is on two knees or feet and one hand, with the other hand used to interact with an object.
	crawling	The infant moves on their hands and knees with their stomach off the ground.
Prone	pivoting	The infant is lying prone and rotating or turning in a small area on the floor without moving forward or backward.
	prone	The infant is lying flat on their stomach.
	side lying	The infant is lying on their side with their torso oriented sideways.
	belly crawling	The infant moves forward by sliding their belly along the ground.
Sitting	supported sitting by the caregiver	The infant sits upright on their bottom with their back supported by the caregiver, and their legs either extended in front or folded beneath them.
	supported sitting using own hands as support	The infant is sitting on their bottom, leaning to one side with the support of their arm, and their legs extended in front or to the side.
	independent sitting	The infant sits upright on their bottom with their legs in front or folded beneath them, without leaning on any external support.
Supine	supine	The infant is lying flat on their back.
Upright	supported stand	The infant is standing upright with straight legs and feet on the floor while being supported by the caregiver and not moving.
	standing upright	The infant is standing upright with straight legs and feet on the floor but is not moving.
	walking	The infant is moving upright with legs straight and feet on the floor.
	supported walking	The infant walks upright with legs straight and feet on the floor, supported by the caregiver’s hands.

## Data Availability

The datasets presented in this article will be available upon request from the corresponding authors following an embargo period from the date of publication to allow for the finalisation of the ongoing longitudinal project. The computer code used in this study is openly available on GitHub: https://github.com/aleksanderrogowski/explaining-CatBoost-model-parameters (accessed 7 October 2024).

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
