# Peer review of "Identifying Infant Body Position from Inertial Sensors with Machine Learning: Which Parameters Matter?"

_sensors, 2024, doi:10.3390/s24237809_

Round 1

Reviewer 1 Report

Comments and Suggestions for Authors

Abstract

·         Extra space in line 10 before the apostrophe next to infants

·         SHAP acronym should be defined in the abstract

Introduction

·         The introduction should begin with a discussion of the need (“assessment of physical development and motor skills during infancy”) rather than jumping immediately to the approach the authors are taking. The introduction focuses exclusively on IMU approaches and does not acknowledge the potential value of video (and automated pose estimation from video) and/or the associated challenges. This section should be reframed to introduce the clinical need, the different technological approaches, and then motivate the authors’ approach relative to previous efforts using IMUs.

·         It is unclear to me what gap this paper seeks to fill. The organization of the introduction could be improved to better communicate the position of this work in the field. The authors acknowledge that deep learning has been used to address the problem statement, but it is unclear whether the authors’ study addresses the limitations based on the introduction.

·         The emphasis on feature importance and it’s potential implications for understanding human development is novel.

Materials and Methods

·         Citation for the previous study that collected these data should be cited again in the materials and methods (perhaps in section 2.1), not just in the introduction

·         Table 4 could probably be removed, as the abbreviations are provided in line in the preceding paragraph

·         Figure 6 does not add much information for a publication – the authors are using an established cross validation strategy. Consider this as a supplemental figure.

Results

·         Is it possible to do statistical comparisons of the CatBoost and Random Forest algorithms to support the authors’ statement that it outperforms in every posture?

·         The authors have descriptors such as “acceptable” in the results – on what is this based? The authors should provide rationale for this.

Discussion

·         It seems to me that the omission of video-based approaches from the discussion is a major limitation as the results are not positioned in the context of the literature that should be addressed

Appendices

·         The appendices are separate and should not included in the core manuscript, correct?

Author Response

Abstract

  •     Comments 1: Extra space in line 10 before the apostrophe next to infants
    Response 1: Thank you for this comment, we did find some extra white spaces left in different lines. The formatting in the version downloaded from the Sensor site causes an illusion of extra space in the text in this place.

________________________________________

  •     Comments 2: SHAP acronym should be defined in the abstract
    Response 2: Added.

________________________________________

Introduction

  •     Comments 3: The introduction should begin with a discussion of the need (“assessment of physical development and motor skills during infancy”) rather than jumping immediately to the approach the authors are taking. The introduction focuses exclusively on IMU approaches and does not acknowledge the potential value of video (and automated pose estimation from video) and/or the associated challenges. This section should be reframed to introduce the clinical need, the different technological approaches, and then motivate the authors’ approach relative to previous efforts using IMUs.

Response 3: Added to the text.

________________________________________

  •     Comments 4: It is unclear to me what gap this paper seeks to fill. The organization of the introduction could be improved to better communicate the position of this work in the field. The authors acknowledge that deep learning has been used to address the problem statement, but it is unclear whether the authors’ study addresses the limitations based on the introduction.

Response 4: Thank you for this comment. Aside from review articles, there appears to be no specific study that directly compares how the number of sensors, the type of machine learning algorithms, and the choice of features affect classification outcomes in wearable devices used for infant monitoring on a single dataset. Most existing research focuses on developing specific algorithms or wearable systems for certain applications, like monitoring vital signs or movement in infants, rather than systematically comparing these factors. This information is crucial for designing an effective research setup.

________________________________________

  •     Comments 5: The emphasis on feature importance and it’s potential implications for understanding human development is novel.

Response 5: Thank you for this comment.

________________________________________

Materials and Methods

  •     Comments 6: Citation for the previous study that collected these data should be cited again in the materials and methods (perhaps in section 2.1), not just in the introduction

Response 6:  Added.

________________________________________

  •     Comments 7: Table 4 could probably be removed, as the abbreviations are provided in line in the preceding paragraph

Response 7: Removed. All abbreviations are explained in the preceding paragraph.

________________________________________

  •     Comments 8: Figure 6 does not add much information for a publication – the authors are using an established cross validation strategy. Consider this as a supplemental figure.

Response 8: Moved to Appendix B.

________________________________________

Results

  •     Comments 9: Is it possible to do statistical comparisons of the CatBoost and Random Forest algorithms to support the authors’ statement that it outperforms in every posture?

Response 9: Thank you for this comment. A new Figure 6 has been added, based on Figure A2, along with the Friedman test and FDR correction.

________________________________________

  •     Comments 10: The authors have descriptors such as “acceptable” in the results – on what is this based? The authors should provide rationale for this.

Response 10:  The criteria for acceptability depend on the specific application and the goals set by the researcher. As such, we do not claim that the observed changes are “acceptable”. We would like to highlight that the largest drop in the F1 score occurs when the Legs sensors are excluded, especially for the Upright position.  

________________________________________

Discussion

  •     Comments 11: It seems to me that the omission of video-based approaches from the discussion is a major limitation as the results are not positioned in the context of the literature that should be addressed

Response 11: Added to the text.

________________________________________

Appendices

  •     Comments 12: The appendices are separate and should not included in the core manuscript, correct?

Response 12: Appendiecies A and B  were present in the main text because it was included in the template from MDPI.

Reviewer 2 Report

Comments and Suggestions for Authors

The paper “Identifying Infant Body Position from Inertial Sensors with Machine Learning: Which Parameters Matter?” presents a comprehensive study on classifying infant and caregiver activities by analyzing data from Inertial Measurement Units (IMUs) attached to infants and caregivers. The authors focus on distinguishing between static and dynamic activities using two machine learning classifiers: the Random Forest Classifier (chosen as a baseline reference) and the CatBoost Classifier, which is known for its handling of categorical data and robustness with small datasets.

The Materials and Methods section is well-detailed, providing clear insight into the experimental setup, data collection process, and data preprocessing steps. The paper presents a critical evaluation of feature contribution using SHAP values, which offer insights into which features significantly impact model predictions. This approach allows for a deeper understanding of how various sensor-derived features, such as acceleration, orientation, and angular velocity, contribute to the classification of static versus dynamic activities.

The Result section provides an in-depth discussion of the classification outcomes. The authors analyze model performance metrics, including accuracy, F1 score, and precision-recall balance, to assess the classifiers’ effectiveness in distinguishing between the two activity types. They further break down the results to examine the classifiers' performance across different activity contexts and discuss potential challenges in separating similar activity patterns. The discussion highlights the importance of particular features and sensor placement in enhancing classification accuracy.

Additionally, the use of SHAP values in feature analysis adds a layer of interpretability, allowing the authors to identify the most influential features for each activity class. This analysis is valuable for future research in wearable-based infant monitoring, as it provides insights into which sensor parameters and positions are most critical for reliable activity recognition.

Suggestion:

While the paper is largely well-written, it would benefit from a careful revision to address minor language errors, typos, and redundant content. For example, section 2.3.1 contains repetitive explanations that could be streamlined for clarity. A thorough proofreading would enhance readability and flow, ensuring the study’s methodology and findings are communicated effectively.

Author Response

Comments 1: While the paper is largely well-written, it would benefit from a careful revision to address minor language errors, typos, and redundant content. For example, section 2.3.1 contains repetitive explanations that could be streamlined for clarity. A thorough proofreading would enhance readability and flow, ensuring the study’s methodology and findings are communicated effectively.

Response 1: Thank you for this comment. Minor changes and improvements have been made throughout the article including section 2.3.1.

Round 2

Reviewer 1 Report

Comments and Suggestions for Authors

This revision is responsive to the previous review - thank you for considering the feedback.